

# Change in Antarctic Ice Shelf Area from 2009 to 2019

Julia R. Andreasen[1,2], Anna E. Hogg[2], Heather L. Selley[2]

[1]Department of Soil, Water and Climate, University of Minnesota, St. Paul, 55108, USA
[2]School of Earth and Environment, University of Leeds, Leeds LS2 9JT, UK

*Correspondence to*: Julia R. Andreasen (andr0856@umn.edu)

**Abstract.** Antarctic Ice Shelves provide buttressing support to the ice sheet, stabilising the flow of grounded ice and its contribution to global sea levels. Over the past 50-years satellite observations have shown ice shelves collapse, thin and retreat, however, there are few measurements of the Antarctic wide change in ice shelf area. Here, we use MODIS satellite data to measure the change in ice shelf calving front position and area on 34 ice shelves in Antarctica, from 2009 to 2019.

Over the last decade, a reduction in area on the Antarctic Peninsula (6,692.5 km$^2$) and West Antarctica (5,563.1 km$^2$), has been outweighed by area growth in East Antarctica (3,532.1 km$^2$) and the large Ross and Ronne-Filchner Ice Shelves (14,027.9 km$^2$). The largest retreat was observed on Larsen-C Ice Shelf where 5,916.6 km$^2$ of ice was lost during an individual calving event in 2017, and the largest area increase was observed on Ronne Ice Shelf in East Antarctica, where gradual advance over the past decade (535.3 km$^2$/yr) led to a 5,888.6 km$^2$ area gain from 2009-2019. Overall, the Antarctic

Ice Shelf area has grown by 5,304.5 km$^2$ since 2009, with 18 ice shelves retreating and 16 larger shelves growing in area. Our observations show that Antarctic Ice Shelves gained 660.6 Gt of ice mass over the decade whereas the steady state approach would estimate substantial ice loss over the same period, demonstrating the importance of using time-variable calving flux observations to measure change.

## 1 Introduction

Ice shelves fringe three quarters of the Antarctic coastline, providing buttressing support to the grounded ice and linking the ice sheet with the Southern Ocean. The calving front represents the seaward limit of the ice shelf edge and is the boundary of the Antarctic coastal margin. The calving front location (CFL) can change gradually through sustained growth or retreat (Cook and Vaughan, 2010) or more suddenly due to large events such as iceberg calving (Hogg and Gudmundsson, 2017) and ice shelf collapse (Rott et al., 1996; Rack and Rott, 2004; Padman et al., 2012). Mapping the time-variable calving front

location on Antarctic Ice Shelves is important for i) estimating the total ice shelf freshwater budget, ii) as a precursor for dynamic instability and therefore ice sheet sea level contribution, iii) as an indicator of changing ice shelf structural conditions, and iv) as a proxy for changing ocean and atmospheric forcing. Satellite observations have shown that a reduction in ice shelf area can cause upstream glaciers to thin (Scambos et al., 2004) and accelerate by up to eight times their previous speed (Rignot et al., 2004), increasing the ice dynamic sea level contribution from the affected region. Some zones

of floating ice provide significantly more structural stability to the ice sheet, with ice inland of the compressive arch or in



contact with a pinning point, triggering instability if lost (Holland et al., 2015). The effect of change in ice shelf area is not always local, with studies showing that ice shelves provide far-reaching buttressing support to grounded ice hundreds of kilometres away (Fürst et al., 2016). However, many iceberg calving events form part of the natural cycle of ice shelf evolution, with steady regrowth and advance of the calving front typically seen after a calving event (Hogg and Gudmundsson, 2017).

Over the past 30-years, ice shelves across Antarctica have been observed to advance steadily, retreat after iceberg calving events, and collapse catastrophically as seen in the case of the Larsen-A (Rott et al., 1996), Larsen-B (Rack and Rott, 2004) and Wilkins Ice Shelves (Padman et al., 2012) on the Antarctic Peninsula. Tracking change in the calving front location is a vital input parameter for ice flow models, as it is used to inform studies of calving processes and their driving forces (Trevers et al., 2019), and is required to compute ice shelf mass change from calving, a component part of the total budget along with basal melt and surface mass input (Rignot et al., 2013). Measurements of the ice shelf calving front location have been made using a range of methods, including historical ship-based observations dating from 1842 on the Ross Ice Shelf (Jacobs et al., 1986; Keys et al., 1998), manual delineation of images acquired by aerial photography (Cook et al., 2005) and optical and Synthetic Aperture Radar (SAR) satellites (Cook and Vaughan, 2010; MacGregor et al., 2012), automated ice front detection (Baumhoer et al., 2019) and by applying edge detection techniques to satellite radar altimetry elevation data (Wuite et al., 2019). The spatial resolution, accuracy and frequency of these complimentary techniques vary, with the temporal and spatial extent of calving front measurements largely dependent on the repeat period and coverage of data acquired, and the manual intensity of the processing technique used. While data prior to the satellite era (pre-1960's) is extremely limited, historical records are an important reference dataset for understanding long-term change in the ice front position and its response to environmental forcing. Despite the importance of this glaciological parameter, circum-Antarctic surveys of the change in ice shelf calving front position are rare (Baumhoer et al., 2018, 2019; Greene et al., 2022), and there are only five examples of regional assessments that have been updated since 2011 (MacGregor et al., 2012; Wuite et al., 2019; Lilien et al., 2018; Francis et al., 2021; Greene et al., 2022). In this study we address this gap by mapping the annual calving front location on 34 ice shelves around Antarctica from 2009 to 2019, using MODIS satellite imagery (Scambos et al., 1996). The results provide a comprehensive assessment of ice front migration across Antarctica over the last decade, enabling areas of growth and retreat to be accurately quantified (Fig. 1).

## 2 Data and Methods

We measured the annual calving front position on 34 ice shelves, encompassing 80% of the Antarctic coastline, over the 11-years from 2009 to 2019 (Fig. 1). We used over 350 multispectral optical images, acquired by the Moderate Resolution Imaging Spectroradiometer (MODIS) instrument onboard the NASA Terra and Aqua satellites (Scambos et al., 1996) (Table S1). Images acquired during the austral summer, from mid-January to the end of February, were selected throughout the decade to ensure consistent sampling and to avoid aliasing seasonal variation in the calving front position. Cloud free



satellite images with open ocean at the calving front were preferentially selected whenever possible, as the presence of sea ice and iceberg melange can reduce the accuracy with which the calving front location can be visually identified. Images

acquired around midday were also prioritised as the illumination at this time provides better contrast, enabling clearer identification of the ice shelf edge. The study period started in 2009 on 30 ice shelves, however, on the Wordie, Baudouin, Nansen and Drygalski Ice Shelves suitable images were not acquired until 2011; therefore, this was used as the earliest start date in these three regions throughout this study (Table 1). We produced an annual measurement of the ice front position on 34 ice shelves around Antarctica, by manually delineating the calving front location at the point where the ice shelf surface

visibly transitioned to open ocean or sea ice, in each satellite image (Cook et al., 2005; Cook and Vaughan, 2010).

**Figure 1: Antarctic map of ice shelf area change from 2009 to 2019 with ice shelf names overlaid on a Bedmap2 surface of**
**Antarctica. Circle areas denote total amount of ice shelf area in km² lost (red) or gained (blue). The black bold line represents the Antarctic coastline, combining 2015 and 2019 data.**



Equidistant points were plotted every ~1000 meters along the ice front using a Polar Stereographic projection, to standardize the distance scale, point density, and accuracy of the calving front boundary on every ice shelf. The ice shelf calving front

location is constantly evolving with the measured location representative of the time stamp of the satellite image used, however, for the purposes of this study we assume this reflects the annual location. The accuracy of the calving front position is limited by the georeferencing precision of the image, and the digitisation of the boundary. We assessed the uncertainty in the measurement technique by delineating the calving front boundary five times using the 2017 image on the Dotson Ice Shelf, and then measuring the variance from the mean position. The results show that the standard deviation of

the calving front measurement is 254 meters which we assume to be our measurement uncertainty. This reflects the spatial resolution of the MODIS imagery which has a pixel size of 250 x 250 meters. The contemporary calving front positions from this study were combined with historical measurements on the Antarctic Peninsula to extend the record of change back to 1947, including the Larsen-A to -C, George VI, Wilkins, Wordie, Bach, and Stange Ice Shelves (Cook and Vaughan, 2010). Overall, this study has produced 366 calving front measurements between 2009 to 2019, and utilised 53 historical

measurements on the Antarctic Peninsula, to provide the most temporally and spatially extensive assessment of change in ice front position across Antarctica.

The annual area of each ice shelf was measured from 2009 to 2019 by combining the digitized calving front locations with a reference grounding line position, which marks the inland limit of the ice shelf boundary (Thomas et al., 1979). The grounding line and calving front positions were polygonised and then intersected for each Antarctic drainage basin fringed

by an ice shelf, with isolated islands and nunataks subtracted from the area, creating a bounded area for each ice shelf for each year of the study. The total area change over the decade long study period was calculated by differencing the most recent ice shelf area observation (2019) from the oldest (2009 or 2011) (Table 1). We computed the mean annual rate of calving by dividing the total area change by the number of years observed and calculated the percentage area change by dividing the total area change by the 2009 area (Table 1). To assess the volume and ice mass change caused by calving front

evolution, we extracted ice thickness from Bedmap2 (Fretwell et al., 2013) across the most inland measured calving front position, which ranged from 2009 to 2019 depending on the ice shelf (Table 1), calculating a mean thickness at the calving front for each shelf. We calculated the annual mass change of each ice shelf due to calving processes by computing the volume change of the calved ice and multiplying each annual area by the mean ice thickness and ice density (0.9166 Gt/km$^3$). The mean rate of volume change was computed by dividing the annually varying ice shelf volume change by the

study period. As the accuracy of ice shelf area measurements depend on both variations in the width and the length of the coastline, we rounded this to 1 km$^2$ precision in line with previous studies (Cook and Vaughan, 2010). The same method of calculating the area, volume and calving mass change was applied to the historical calving front positions on the Antarctic Peninsula.




| Ice Shelf | First-Last Dates (yr) | Years Observed (yr) | First Recorded Area (km²) | Last Recorded Area (km²) | Area Difference 2009-19 (km²) | Area Percentage Change (%) | Rate of Area Change (km²/yr) | Year of Max Area Change (yr) | Max Area Change (km²) | Most Inland Calving Front (yr) | Mean Ice Thickness (km/yr) | Mean Ice Speed (km/yr) | Inland CFL Length (km) | Observed Calving Flux (Gt) | Steady State Calving Flux (Gt) |
|---|---|---|---|---|---|---|---|---|---|---|---|---|---|---|---|
| Larsen A | 2009-19 | 11 | 32.4 | 21.9 | -10.5 | -32.5% | -1 | 2011 | -4.9 | 2019 | 0.15 | 0.03 | 8.8 | -1.4 | -0.4 |
| Larsen B | 2009-19 | 11 | 1,850.1 | 2,099.8 | 249.7 | 13.5% | 22.7 | 2014 | 34.6 | 2009 | 0.25 | 0.51 | 48.4 | 57.6 | -62.8 |
| Larsen C | 2009-19 | 11 | 46,548.4 | 41,785.6 | -4,762.7 | -10.2% | -433.0 | 2017 | -5,916.6 | 2018 | 0.22 | 0.51 | 348.2 | -976 | -397 |
| George | 2009-19 | 11 | 23,484.4 | 22,882.4 | -601.9 | -2.6% | -54.7 | 2010 | -212.7 | 2017 | 0.17 | 0.35 | 315.1 | -92.8 | -185.6 |
| Wilkins | 2009-19 | 11 | 11,406.4 | 10,202.3 | -1,204.1 | -10.6% | -109.5 | 2009 | -780.1 | 2019 | 0.06 | 0.11 | 279.6 | -64.3 | -17.6 |
| Wordie | 2011-19 | 9 | 87.1 | 47.7 | -39.4 | -45.2% | -4.4 | 2013 | -34.7 | 2017 | 0.11 | 0.74 | 37.2 | -4.1 | -25.5 |
| Bach | 2009-19 | 11 | 4,553.1 | 4,439.8 | -113.3 | -2.5% | -10.3 | 2017 | -42.6 | 2018 | 0.22 | 0.11 | 49.1 | -22.3 | -11.4 |
| Stange | 2009-19 | 11 | 8,016.6 | 7,806.3 | -210.3 | -2.6% | -19.1 | 2014 | -86.1 | 2019 | 0.18 | 0.23 | 301.8 | -34.2 | -122.1 |
| **AP Total** | - | **10.8** | **95,978.4** | **89,285.9** | **-6,692.5** | **-7.0%** | - | - | - | - | **0.17** | **0.32** | **1,388.13** | **-1,137.4** | **-822.5** |
| Swinburn | 2009-19 | 11 | 905.2 | 720.2 | -185 | -20.4% | -16.8 | 2016 | -43.8 | 2019 | 0.26 | 0.14 | 75.2 | -44.6 | -27.5 |
| Sulzberger | 2009-19 | 11 | 12,489.6 | 12,275.7 | -213.8 | -1.7% | -19.4 | 2011 | -142.1 | 2019 | 0.21 | 0.13 | 312.7 | -41.4 | -86.7 |
| Getz East | 2009-19 | 11 | 14,627.4 | 14,405.3 | -222.1 | -1.5% | -20.2 | 2017 | -87.5 | 2018 | 0.32 | 0.28 | 190.1 | -64.4 | -167.8 |
| Getz West | 2009-19 | 11 | 19,381 | 19,199.9 | -181.1 | -0.9% | -16.5 | 2015 | -237.5 | 2018 | 0.30 | 0.62 | 298.5 | -49.3 | -555.2 |
| Dotson | 2009-19 | 11 | 5,791.3 | 5,843.3 | 52 | 0.9% | 4.7 | 2016 | -51.8 | 2009 | 0.34 | 0.35 | 50.4 | 16.4 | -60.5 |
| Crosson | 2009-19 | 11 | 3,545.2 | 3,840 | 294.8 | 8.3% | 26.8 | 2014 | 195.8 | 2014 | 0.31 | 1.11 | 391.7 | 83.3 | -1,352.8 |
| Thwaites | 2009-19 | 11 | 5,448.9 | 2,524.8 | -2,924.1 | -53.7% | -265.8 | 2012 | -2,786.3 | 2017 | 0.40 | 1.72 | 421.7 | -1,074 | -2,928.2 |
| Pine Island | 2009-19 | 11 | 6,263.6 | 5,220.9 | -1,042.7 | -16.6% | -94.8 | 2013 | -688.5 | 2019 | 0.32 | 0.91 | 202.8 | -309.6 | -604.6 |
| Abbot | 2009-19 | 11 | 31,826.8 | 30,685.9 | -1,140.9 | -3.6% | -103.7 | 2010 | -309 | 2016 | 0.16 | 0.12 | 515.2 | -168.2 | -98.1 |
| **WAIS Total** | - | **11** | **100,279** | **94,715.9** | **-5,563.1** | **-5.5%** | - | - | - | - | **0.29** | **0.60** | **2,458.3** | **-1,651.8** | **-5,881.4** |
| Brunt | 2009-19 | 11 | 37,180.4 | 39,061.1 | 1,880.6 | 5.1% | 171 | 2014 | 336.1 | 2009 | 0.15 | 0.85 | 892.1 | 251.3 | -1,119 |
| Riiser | 2009-19 | 11 | 43,544.3 | 44,041.8 | 497.5 | 1.1% | 45.2 | 2011 | 111.3 | 2009 | 0.22 | 0.20 | 760.3 | 98.8 | -330 |
| Fimbul | 2009-19 | 11 | 40,801.1 | 41,277.3 | 476.2 | 1.2% | 43.3 | 2013 | 127.4 | 2009 | 0.21 | 0.28 | 1,093.6 | 90.6 | -644.6 |
| Baudouin | 2011-19 | 9 | 33,142.8 | 32,903.4 | -239.4 | -0.7% | -26.6 | 2017 | -174.8 | 2018 | 0.22 | 0.16 | 972.5 | -47.4 | -273.3 |
| Amery | 2009-19 | 11 | 60,797 | 62,299.4 | 1,502.4 | 2.5% | 136.6 | 2013 | 229.7 | 2009 | 0.28 | 0.86 | 661.0 | 386.8 | -1,609.8 |
| West | 2009-19 | 11 | 15,854.8 | 16,342.5 | 487.7 | 3.1% | 44.3 | 2013 | -667.7 | 2014 | 0.20 | 0.42 | 1,032.1 | 90 | -873.5 |
| Shackleton | 2009-19 | 11 | 26,216.5 | 27,057.2 | 840.7 | 3.2% | 76.4 | 2013 | 152.5 | 2009 | 0.20 | 0.59 | 1,464.9 | 150.7 | -1,703.6 |
| Totten | 2009-19 | 11 | 6,109.2 | 5,956.2 | -152.9 | -2.5% | -13.9 | 2015 | -229.7 | 2016 | 0.30 | 1.42 | 125.3 | -42.7 | -545.7 |
| Moscow Uni | 2009-19 | 11 | 5,898.8 | 6,029.8 | 131 | 2.2% | 11.9 | 2015 | -126.2 | 2011 | 0.38 | 0.45 | 278.8 | 45.7 | -486.3 |
| Mertz | 2009-19 | 11 | 5,408.9 | 3,346 | -2,063 | -38.1% | -187.5 | 2010 | -2,450.8 | 2011 | 0.29 | 0.90 | 153.9 | -546.7 | -404.7 |
| Ninnis | 2009-19 | 11 | 1,779.6 | 2,048.1 | 268.4 | 15.1% | 24.4 | 2018 | 88.7 | 2009 | 0.43 | 0.77 | 349.5 | 105.3 | -1,162.6 |
| Nansen | 2011-19 | 9 | 2,011 | 1,865.5 | -145.5 | -7.2% | -16.2 | 2016 | -181.6 | 2018 | 0.14 | 0.25 | 127.5 | -19.1 | -37.5 |
| Drygalski | 2011-19 | 9 | 2,335.3 | 2,383.6 | 48.3 | 2.1% | 5.4 | 2017 | 27.2 | 2013 | 0.21 | 0.67 | 2,44.3 | 9.4 | -287.8 |
| **EAIS Total** | - | **10.5** | **28,1079.7** | **28,4611.7** | **3,532.1** | **1.3%** | - | - | - | - | **0.25** | **0.60** | **8,156.0** | **572.6** | **-11,751.4** |
| Ross East | 2009-19 | 11 | 194,737.7 | 196,789 | 2,051.4 | 1.1% | 186.5 | 2013 | 253.6 | 2009 | 0.10 | 0.77 | 371.5 | 181.7 | -277.6 |
| Ross West | 2009-19 | 11 | 306,983.3 | 310,828.1 | 3,844.9 | 1.3% | 349.5 | 2013 | 475.7 | 2009 | 0.23 | 0.54 | 924.4 | 826.1 | -1,186.6 |
| Ronne | 2009-19 | 11 | 339,124.1 | 345,012.7 | 5,888.6 | 1.7% | 535.3 | 2010 | 650.8 | 2009 | 0.20 | 0.73 | 859.5 | 1,089.8 | -1,273.3 |
| Filchner | 2009-19 | 11 | 103,920.2 | 106,163.4 | 2,243.2 | 2.2% | 203.9 | 2016 | 313.6 | 2009 | 0.38 | 1.11 | 262.1 | 779.6 | -1,108.2 |
| **Large Shelf Total** | - | **11** | **944,765.3** | **958,793.2** | **14,027.9** | **1.5%** | - | - | - | - | **0.23** | **0.79** | **2,417.5** | **2,877.2** | **-3,845.6** |
| **Antarctica Total** | - | **10.8** | **1,422,102.3** | **1,427,406.8** | **5,304.5** | **0.4%** | - | - | - | - | **0.24** | **0.56** | **14,419.93** | **660.6** | **-20,028.1** |

**Table 1: Summary table with data on each ice shelf including: area change from 2009 to 2019, the absolute difference, percentage difference, and rate of change between the first and last recorded dates, the year and amount of maximum area change, the year**



and length of the most inland calving front and the averaged ice thickness and speed at each most inland calving front, and the observed calving flux and steady state calving fluxes.

## 3 Results & Discussion

This study presents a spatially and temporally extensive record of calving front location and area change on 34 major ice shelves in Antarctica, from 2009 to 2019 (Fig. 1; Table 1), with 3 of those shelves measured from 2009 to 2021. Over the 11-years from 2009-2019 we observed six distinct types of ice shelf calving front behaviour, characterized by a) major

calving events, b) rapid calving front retreat, c) gradual calving front retreat, d) growth with periodic retreat, e) steady calving front advance and f) rapid area growth. We grouped the ice shelves into these six categories to describe the changes observed to provide a detailed evaluation of ice shelf behaviour in Antarctica over the last decade.

### 3.1 Major Calving Events

Major calving events are defined as the loss of a significant proportion of the ice shelf, >5% of the total area, resulting in the

production of one or more icebergs over a short time-period (< month). Six ice shelves in Antarctica experienced major calving events between 2009 and 2019, including the Wilkins, Wordie and Larsen-C Ice Shelves on the Antarctic Peninsula in 2009, 2013 and 2017 respectively; Thwaites Glacier in West Antarctica in 2012, and Mertz, and Nansen Ice Shelves in East Antarctica in 2010 and 2016 respectively (Fig. 3a.). Thwaites Glacier Ice Shelf experienced the largest relative area change losing a total of 53.7% (-2,924.1 $km^2$) of its original area (Table 1), due to the combined effects of both iceberg

calving (2012 ice tongue calving event) and retreat (Sup. Fig. 32). Between 1963 and 2008 the Larsen-C Ice Shelf retained 91 % of its area (50,837 $km^2$) (ice fronts provided by Cook and Vaughan, 2010); however, in 2017 it calved a >200 km long iceberg (A68) (Hogg and Gudmundsson, 2017), reducing its area by 12.7 % (-5,916.6 $km^2$) and resulting in an overall ice loss of 10.2 % from 2009-2019 (Sup. Fig. 3).

The Wordie Ice Shelf on the western edge of the Antarctic Peninsula, decreased in area by 90 % between 1966 and 2008 (ice

fronts provided by Cook and Vaughan, 2010), resulting in four isolated ice shelf remnants buttressing the Carlson, Prospect, and Hariot Glaciers and unnamed remnant between Hariot and Fleming Glaciers (Sup. Fig. 6). Several ice rises played a crucial stabilizing role in the Wordie grounding zone; however, they also act as wedges splintering and weakening the ice shelf from the three inflowing tributary glaciers (Vaughan, 1993). Between 2011 and 2019, the Wordie Ice Shelf lost 45.2% of its remaining area (Table 1), with 88% of this loss caused by a 34.7 $km^2$ calving event in 2013. After a period of sustained

retreat since 1990 (Cook and Vaughan, 2010), the north and western portions of Wilkins Ice Shelf retreated by 1,204.1 $km^2$ between 2009 and 2010 (Sup. Fig. 5). This was due to the loss of a 1 km wide ice bridge to Charcot Island, thought to be caused by Easterly winds driving cyclic motion and ice mélange pressurized against the ice bridge due to wind-stress (Humbert et al., 2010).

In 2010, Mertz Glacier calved a 78 km long iceberg (C28) losing 45.3 % of its original area (-2,450.8 $km^2$), after the B-09B

iceberg which calved from the Ross Ice Shelf in 1987 collided with the highly crevassed floating ice tongue (Fig. 2a., Sup.





Fig. 20) (Massom et al., 2015). Grounded icebergs around the Mertz coastline influence the floating shelf by creating a layer of fast-ice cover that extends the ice tongue's length, additionally Mertz is responsible for the drainage of 0.8% of the EAIS (Massom et al., 2015). The Nansen Ice Shelf grew steadily at an average rate of 6.9 km²/yr from 2011-2016, however, between 2016 and 2017 8.9 % (181.6 km²) of the total ice shelf area was lost through calving (Sup. Fig. 22). The fracture

that formed the C-33 and C-33b icebergs was first recorded in 1987, growing at a rate of 6.6 km/yr from 2011 to 2013, (Li et al., 2016; Dziak et al., 2018), with the eventual calving in 2016 thought to be triggered by a low-pressure storm (Dziak et al., 2018). Mertz Glacier and the Nansen Ice Shelf are the only two regions in East Antarctica to have major calving events between 2009 and 2019 and are two of four ice shelves in East Antarctica to experience a net area loss over the 11-year study period. After their calving events, the Wilkins and Thwaites Ice Shelves continued to retreat at a more gradual rate,

whereas the Mertz Ice Shelf re-advanced and the extent of the Wordie Ice Shelf remained relatively static. Between 2019 and 2022, major calving events have occurred on the Amery Ice Shelf (Sep 2019, iceberg D-28, 1636 km², Francis et al., 2021) (Sup. Fig. 15), the Brunt Ice Shelf (Feb 2021, iceberg A-74, 1270 km²) (Sup. Fig. 11), and the Ronne Ice Shelf (May 2021, iceberg A-76, 4,310 km²) (Sup. Fig. 9), significantly expanding the region in East Antarctica that has experienced a net area loss since 2009. Future studies are needed to assess whether there has been a significant increase in iceberg calving over the

past decade, or whether the more frequent repeat period of satellite observations has better captured the true frequency of major iceberg calving events in Antarctica.

**3.2 Rapid Calving Front Retreat**

Rapid calving front retreat is defined as ice shelves that have experience sustained and significant ice loss throughout the 11-year study period, loosing at least 15 % of their total area. Three ice shelves in Antarctica experienced rapid calving front

retreat between 2009 and 2019, including the Larsen-A Ice Shelf on the Antarctic Peninsula and Pine Island Glacier and Swinburne Ice Shelf in West Antarctica (Fig. 3b.). These ice shelves are fed by fast-flowing glaciers, resulting in observations of modest annual advance (typically between 1 % and 2 %). Larsen-A Ice Shelf on the north-east Antarctic Peninsula had a total area of 2,928.9 km² in 1963 and began to experience some collapse in the 1980s (Sup. Fig. 1) (ice fronts provided by Cook and Vaughan, 2010). In January 1995, a collapse of 2,270 km² of ice (Rott et al., 1996), left Larsen-

A with only 681.6 km² of its original area after surface meltwater ponds triggered hydrofracturing through crevasses (Scambos et al., 2000). From 1995 to 2008, Larsen-A experienced steady retreat of the ice shelf remnant with 637 km² of ice lost, and from 2009-2019 the Larsen-A Ice Shelf continued to retreat, losing a further 10.5 km² of ice and leaving a remaining area of 21.9 km². In this study, all Larsen-A area calculations do not include the Seal Nunataks region.

Pine Island Glacier (PIG) is in the Amundsen Sea Embayment where incursions of warm Circumpolar Deep Water (CDW)

onto the continental shelf have caused high basal melt rates (Dutrieux et al., 2014) and accelerating ice discharge into the ocean (Fig. 2b., Sup. Fig. 33) (Joughin et al., 2014; MacGregor et al., 2012). Previous studies have shown that PIG has experienced long-term ice shelf retreat since the 1970s (Crabtree and Doake, 1982), where unbuttressing of the grounded ice has caused large, negative ice dynamic loss from the basin over the past three decades (Mouginot et al., 2014). In 2011, a rift



occurred across the shelf further inland than previously on record (since 1947), causing a calving event in 2013 where 688.5

km$^2$ of ice was lost. Since 2009, our results show that PIG has retreated by 1,042.7 km$^2$, at a rate of -94.8 km$^2$/yr. Elsewhere in West Antarctica, Swinburne Ice Shelf experienced a relatively uniform rate of ice loss throughout the study period, with a total of 185 km$^2$ of ice lost at an average rate of -16.8 km$^2$/yr, losing 20.4% of its area by 2019 (Sup. Fig. 26).





**Figure 2: Maps of calving front change from 1947-2019 representing a. a major calving event b. rapid calving front retreat, c.**
**gradual calving front retreat, d. growth with periodic retreat, e. steady calving front advance growth, and f. rapid area growth overlaying MODIS satellite images from 2019 (Scambos et al., 1996).**

**3.3 Gradual Calving Front Retreat**

We define gradual calving front retreat as ice shelves that lost less than 4 % of their total area over the 11-year study period, where the maximum percentage of annual growth is 1.31 % and the maximum retreat is -3.76 %. This category is the largest
grouping of ice shelves containing eight locations in Antarctica, including George VI, Bach, and Stange Ice Shelves on the Antarctic Peninsula, Sulzberger, Getz and Abbot Ice Shelves in West Antarctica, and Totten and Baudouin Ice Shelves in East Antarctica (Fig. 3c.).

George VI (GVI) Ice Shelf is the largest ice shelf (22,882.4 km$^2$ in 2019) on the Antarctic Peninsula's western coast and exists as an ice-bridged channel between Alexander Island and the continent (Sup. Fig. 4). GVI has two ice fronts 500 km
apart, with the northern front and the southern front facing Marguerite Bay and Belgica Trough respectively. GVI experiences seasonal surface melt in the austral summer and high rates of basal melting which have been attributed to warm southeast Pacific basin water supplied by CDW beneath the shelf (Lucchitta and Rosanova, 1998). From 1947 to 2008, George VI lost 1,943.7 km$^2$ of ice, therefore retaining 92.4% of its 1947 size (ice fronts provided by Cook & Vaughan, 2010). This slow but steady ice loss continued from 2009 until 2019 at an average rate of 54.7 km$^2$/yr with 601.9 km$^2$ of ice
lost throughout the 11-year study period. The Bach and Stange Ice Shelves are located on either side of the George VI southern opening and have also exhibited similar slow and steady rates of retreat over many decades. Bach lost 303.6 km$^2$ of ice over a 62-year period from 1947 to 2008 (ice fronts provided by Cook & Vaughan, 2010), with a further 113.3 km$^2$ (2.5 %) ice lost in the 11-year period between 2009 and 2019 (Fig. 2c., Sup. Fig. 7). Stange Ice Shelf exhibited similar behaviour, retaining 97 % of its originally recorded ice area between 1973 and 2008, (a loss of 271.9 km$^2$), and losing a further 2.6 % of
its area (210.3 km$^2$) in the last decade (Sup. Fig. 8).

The Getz and Abbot Ice Shelves are the largest shelves in West Antarctica, flanking large portions of the coastline. These ice shelves are anchored at their calving front by a series of islands which limit the rate and scale of iceberg calving events. While grounded ice inland of the Getz Ice Shelf has exhibited ice dynamic speedup over the last 20-years (Selley et al., 2021), the ice shelf area (33,605.2 km$^2$ in 2019) has remained relatively stable. Overall, 403.2 km$^2$ of ice was lost from the
ice shelf over the past decade, with a calving front retreat rate of 16.5 km$^2$/yr in the western portion (Sup. Fig. 28) and 20.2 km$^2$/yr on the eastern shelf (Sup. Fig. 29). During the 11-year study period from 2009 and 2019, both ice shelves have lost a small amount of their total area. Abbot Ice Shelf lost 3.6 % of its total area (1,140.9 km$^2$), with some periods of modest growth in 2014, and from 2016 to 2019 (Sup. Fig. 34). Sulzberger Ice Shelf is located in West Antarctica between Swinburne Ice Shelf and the Guest Peninsula facing the Ross Sea, with an area of 12,275.7 km$^2$ in 2019 (Sup. Fig. 27).
Sulzberger has a complex structure with numerous islands and pinning points flanking the ice front, with ice that is less than 80 m thick on average at its terminus and an ocean depth of ~150 m (Le Brocq et al., 2010). Satellite observations indicate that there have been no significant ice dynamic speed changes on this ice shelf over the past 35-years (Brunt et al., 2011),





however, the pattern of ice flow around the eleven ice rises (and smaller ice rumples) has created lines of weakness which

may increase the likelihood of ice fracturing (Matsuoka et al., 2015). In 2011, a Japanese earthquake triggered a Tsunami

that caused rifts to form on Sulzberger Ice Shelf. This directly led to a 10 km by 6 km iceberg calving event (Brunt et al.,

2011) that reduced Sulzberger's area by 142.1 km$^2$. Between 2009 and 2019 Sulzberger overall retreated at a gradual rate of

19.4 km$^2$/yr losing 1.7 % of its total area.

In East Antarctica, Baudouin Ice Shelf is located on the northern coast and experienced an overall area loss of 239.4 km$^2$

between 2011 and 2019, with an average retreat of 26.6 km$^2$/yr (Sup. Fig. 14). Totten Glacier is also in East Antarctica and

drains the large Aurora Subglacial Basin, which contains enough ice to raise global sea levels by 3.5 m (Sup. Fig. 18)

(Greenbaum et al., 2015). Ice flux from Totten Glacier is the largest in EAIS and third highest behind Pine Island and

Thwaites in Antarctica (Roberts et al., 2018), and the ice streams deeply grounded bed geometry makes the region

susceptible to grounding line retreat and MISI. Between 2009 and 2019 our results show that Totten Glacier Ice Shelf

experienced a total ice loss of 152.9 km$^2$ at an average rate of -13.9 km$^2$/yr. By the end of the study period in 2019 Totten

Glacier Ice Shelf retained 97.5% of its 2009 area.

### 3.4 Growth with Periodic Retreat

We define the growth with periodic retreat category as ice shelves that have faced overall growth of at least 0.9 %, but also

have individual years of retreat within the last decade that range from -0.02 % to -4.21 %. Since 2009, this category includes

Dotson Ice Shelf in West Antarctica, and West, Moscow University and Drygalski Ice Shelves in East Antarctica. The

calving events observed over the last 11-years on these ice shelves are often small in size, with area re-growth occurring in

subsequent years (Fig. 3d.).

Dotson Ice Shelf had an area of 5,791.3 km$^2$ in 2009 and is located in the Amundsen Sea Embayment between the larger

Thwaites and Getz Ice Shelves (Sup. Fig. 30). Over the last 11-years Dotson Ice Shelf's area has grown modestly by 0.9 %,

to 5,843.3 km$^2$. Ice flow from Kohler Glacier is the primary input driving this advance, however, in 2016 a small calving

event occurred, causing a total of 51.8 km$^2$ of ice to be lost. Subsequent re-growth from 2017 to 2019 led to an area gain of

9.9 km$^2$ over the following two years. West Ice Shelf has a large area of 15,854.8 km$^2$ in 2009 and is located along the coast

of the EAIS between the Amery and Shackleton Ice Shelves (Fig. 2d., Sup. Fig. 16). Our results showed that by 2019 the ice

shelf area had increased moderately by 3.1 % to 16,342.5 km$^2$, with one small iceberg calving event in 2013, losing 667.7

km$^2$ of ice. Moscow University Ice Shelf experienced three years of calving front retreat in 2010, 2015, and 2016, totalling

251.3 km$^2$ of ice loss, but overall Moscow University witnessed an overall marginal growth of 131 km$^2$ from 2009 to 2019

(Sup. Fig. 19). The 87.8 km (2019) long Drygalski ice tongue is located on the Scott Coast in East Antarctica, adjacent to

Nansen Ice Shelf (Sup. Fig. 23). This ice tongue has a 2019 area of 2,383.6 km$^2$, with ice flow supplied from the David

Glacier which drives its average rate of advance of 5.4 km$^2$/yr. Between 2011 and 2019 the Drygalski ice tongue area grew

by 48.3 km$^2$, with a small iceberg calving event occurring in 2011, and additional retreat in 2012, 2014, and 2016. While





relatively small in comparison to major iceberg calving events, this category of ice shelves demonstrates the importance of making annual calving front measurements to accurately capture the real ice mass loss through calving events.

**Figure 3: Area Percentage change since 2009 a. major calving events b. rapid calving front retreat, c. gradual calving front retreat,**
**d. growth with periodic retreat, e. steady calving front advance, and f. rapid area growth. AP ice shelves represented in blue,**
**WAIS ice shelves represented in orange, and EAIS ice shelves represented in purple.**



### 3.5 Steady Calving Front Advance

We define steady calving front advance as ice shelves that have gradually grown in area, as controlled by the rate of ice flow.
Overall, we find that ice shelves in this category grew on average by just under 4 % over the decade-long study period, with yearly retreat limited to a maximum of 0.29 %, annual growth ranging from 0 to 0.58 %, and with average annual growth of 0.18 %. Ice shelves in this category include the four largest ice shelves in Antarctica, all over 100,000 km$^2$ in area, which tend to contain some of the thickest floating ice. All 8 ice shelves in this category are located in East Antarctica, and include the Ross East and West Ice Shelves, Ronne, Filchner, Riiser-Larsen, Fimbul, Amery, and Shackleton Ice Shelves (Fig. 3e.).

Ross is the largest ice shelf in Antarctica, bridging the gap between the Siple coast on the West Antarctic Ice Sheet and the transantarctic mountains in the east. Over the decade from 2009 to 2019, both the Ross East and Ross West Ice Shelves have grown steadily by a total of 5,896.2 km$^2$, at rates of 186.5 km$^2$/yr and 349.5 km$^2$/yr, respectively (Sup. Fig. 24 and Sup. Fig. 25). Central regions of the Ross Ice Shelf experience periodic thickening, thought to be due to marine ice re-freezing onto the ice shelf base during the austral winter (Adusumilli et al., 2020; Hogg et al., 2021). During the warmer summer months
observations have shown that localised thinning occurs at the eastern ice shelf calving front driven by atmospherically heated Antarctic surface water (Tinto et al., 2019). Thermal heating of surface water in the Ross Sea occurs when strong offshore winds prevent sea-ice from forming, a process which persistently occurs in the Ross Sea Polynya (Lazzara et al., 2008). Prior to 2009, major pre-existing rifts resulted in significant calving events in 1987 and 2008 (Lazzara et al., 2008), with periods of steady ice shelf re-growth in-between. Our observations combined with those from previous studies (Smethie and Jacobs,
2005; Lazzara et al., 2008), suggests that Ross Ice Shelf exists in a regenerative cycle, where multiple decades of growth result in periodic large calving events.

Ronne and Filchner Ice Shelves are located on the opposite side of Antarctica, flowing into the Weddell Sea. From 2009 to 2019, the area of Ronne and Filchner ice shelves increased by 5,888.6 km$^2$ (1.7 %) and 2,243.2 km$^2$ (2.2 %) (Fig. 2e.) respectively, with annual growth rates of 535.3 km$^2$/yr and 203.9 km$^2$/yr (Sup. Fig. 9 and Sup. Fig. 10). Prior to 2009, major
calving events occurred on Filchner Ice Shelf in 1986 where a total area of 11,500 km$^2$ was lost, and 1998 when a 150 by 35 km iceberg calved, returning the ice shelf calving front to its 1947 position (Ferrigno and Gould, 1987). Prior to these calving events, large 19 km wide rifts grew parallel to the ice front from 1957 onwards, illustrating the long-term gradual build-up to the calving events (Swithinbank et al., 1988). In May 2021 Ronne Ice Shelf calved an iceberg from its western edge with an area of 4,310 km$^2$, resulting in a total ice shelf area of 341,957.2 km$^2$.

Since 2009, the area of the Riiser-Larsen Ice Shelf has grown by 1.1 % from 43,544.3 km$^2$ in 2009 to 44,041.8 km$^2$ in 2019, with an average growth rate of 45.2 km$^2$/yr (Sup. Fig. 12). Few historical calving events have been reported on the Riiser-Larsen Ice Shelf, suggesting it has been remained in a relatively stable configuration, with faster flowing ice feeding the southern zone (73°S to 74°S) and slower flowing ice feeding into the north (72°S to 73°S) (Lange and Kohnen, 1985). Fimbul Ice shelf is located adjacent to the Riiser-Larsen in Donning Maud Land and has grown by 1.2 % from 40,801.1 km$^2$





in 2009 and 41,277.3 km$^2$ in 2019 (Sup. Fig. 13). The central region of the Fimbul Ice Shelf is primarily fed by the flow of
        ice from Jutulstraumen Glacier, which flows at speeds of approximately 760 m/a (Neckel et al., 2021). Jutulstraumen Glacier
        splits the ice shelf into the fast-flowing eastern region which includes the Trolltunga ice tongue, and the slower western
        sector (Humbert and Steinhage, 2011).

        The Amery Ice Shelf is located in central East Antarctica and has grown steadily by 1,502.4 km$^2$ from 2009 to 2019. Prior to
2009, Amery's most recent major calving event occurred in 1963/1964, where approximately 10,000 km$^2$ of ice was lost
        (Fricker et al., 2002). At its 2019 average rate of advance, 136.6 km$^2$/yr, it would take an additional 5 to10 years for the
        calving front location to return to its 1960 pre-calved position, possibly indicating a calving cycle of around 60 to 70 years
        (Fricker et al., 2002). Satellite observations have shown several prominent and growing rifts at the centre of the ice shelf
        calving front, and over the past 34+ years this rift has stretched inland and split into two separate branches caused by
transverse ice spreading (Fricker et al., 2002). In early 2019, the northern branch reached a total length of 35 km from the
        main rift, while the southern branch reached 25 km. In September 2019 the western side of this fracture calved off forming a
        30 km wide and 60 km long iceberg (D-28), with this calving event thought to be triggered by major twin polar cyclones
        producing increased tides and winds (Sup. Fig. 15) (Francis et al., 2021). Shackleton Ice Shelf, situated between West Ice
        Shelf and Law Dome, and one of the larger ice shelves in East Antarctica, experienced an overall growth of 840.7 km$^2$ from
305    2009 to 2019 (Sup. Fig. 17).

### 3.6 Rapid Area Growth

        Ice shelves that have undergone rapid area growth increased their size by over 5 % during the 11-year study period, with
        maximum growth of just under 15 %. As with the gradual ice shelf area growth category, calving front advance is controlled
        by the rate of ice flow, however, rapid area growth ice shelves are on average ten times smaller (11,762 km$^2$ rather than
310    141,684 km$^2$) therefore the calving front advance represents a larger proportion of the total area change. Ice shelves in this
        category include the Larsen-B remnant (65°300 S, 61°W) which is advancing after much of the ice shelf collapsed in March
        2002 (Rack and Rott, 2004), as well as Ninnis and Brunt Ice Shelves on the EAIS, and Crosson Ice Shelf along WAIS (Fig.
        3f.).

        The Larsen-B Ice Shelf is located on the north-eastern side of the Antarctic Peninsula. Between 1963 and 2009 the Larsen-B
Ice Shelf lost 83.0 % (9,054.9 km$^2$) of its area, resulting in an 1,850.1 km$^2$ ice shelf remnant in the Scar Inlet in 2009 (Sup.
        Fig. 2) (ice fronts provided by Cook and Vaughan, 2010). The majority of this ice loss occurred during a catastrophic
        collapse event in 2002 where crevasse hydrofracture caused 3,250 km$^2$ of ice to be lost within a few days (Rack and Rott,
        2004; Cook and Vaughan, 2010) due to an original large tabular calving event that occurred in 1995 (Kulessa et al., 2014).
        Observations have shown that area loss from the Larsen-B Ice Shelf has led to unbuttressing of grounded ice on the Antarctic
Peninsula, resulting in an eightfold acceleration in ice flow between 2000 and 2003 and a corresponding increase in sea level
        contribution from this region (Rignot et al., 2004). Over the 11-year period between 2009 and 2019, ice flow into the



remaining portion of the Larsen-B Ice Shelf caused rapid area growth at a rate of 22.7 km²/yr, with a total area gain of 13.5 % (249.7 km²).

Ninnis Ice Shelf is located next to the Mertz Glacier tongue on the George V Coast of East Antarctica and grew by 268.4
km² (15.1 %) since 2009 (Fig. 2f., Sup. Fig. 21). Ice shelf growth was not uniform across the full extent of Ninnis ice tongue, with a small calving event occurring in 2017 on its eastern side. Brunt Ice Shelf is located to the east of the of Ronne-Filcher Ice Shelf in Dronning Maud Land, with its previous historical calving event occurring 51 years ago in 1971 (Anderson et al., 2013). From 2009 to 2019 the Brunt Ice Shelf grew in area by 1,880.6 km² in total, at a rate of 171 km²/yr (Sup. Fig. 11). In the last decade three major fractures have become active and grown on the Brunt Ice shelf. Chasm 1 is located on the
western side of the Brunt Ice Shelf and lay dormant for 35 years, however satellite observations showed that the crack began to advance in 2012. Chasm 1 propagated across the ice shelf even more rapidly from 2014 onwards, reaching a length of 55 km by 2019. As of 2021, a ~5km long ice bridge connects the tip of Chasm 1 with the McDonald Ice Rumples, a pinning point on the Brunt Ice Shelf. On 31st October 2016, a second fracture called the Halloween Crack was observed on the Brunt Ice Shelf and is over 60 km long growing inland away from the McDonald Ice Rumples. In November 2020 a third crevasse
called the Northern Rift was identified on the eastern side of the McDonald Ice rumples. This crevasse propagated rapidly across the ice shelf and calved a 56 km long, 33 km wide iceberg (A74) in February 2021, resulting in a total area of 38,174.9 km² in 2021. Crosson Ice shelf located adjacent to Dotson Ice Shelf and fed by Pope Glacier and the eastern branch of Smith Glacier, experienced overall rapid area growth of 8.3 % (294.8 km²), with periodic years of retreat in 2010, 2012, 2013, and 2015 (Sup. Fig. 31) (Lilien et al., 2018).

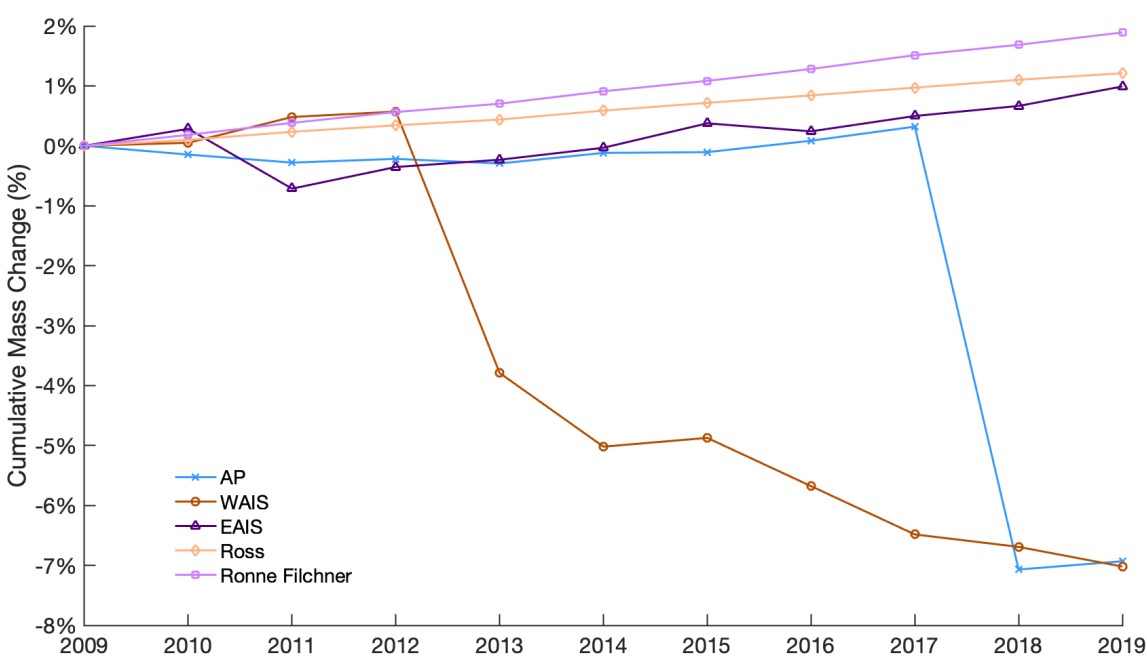




**Figure 4. Cumulative ice shelf mass change from 2009 to 2019 for the AP, WAIS, EAIS, Ross Ice Shelf, and Ronne Filchner Ice Shelves.**

### 3.7 Overall Change in Antarctic Ice Shelf Area

Our results show that over the 11-years from 2009 to 2019, ice shelves in Antarctica gained a modest 0.4 %, or 5,304.5 km$^2$

of their total ice area (Table 1, Fig. 1). This area gain was dominated by significant 14,027.9 km$^2$ (1.5 %) ice shelf area gains on the two largest Antarctic ice shelves, Ronne-Filchner and Ross, and a 3,532.1 km$^2$ (1.3 %) area gain on East Antarctic ice shelves. This counteracted the large reduction in ice shelf area on the Antarctic Peninsula where 7.0 % (-6,692.5 km$^2$) of ice was lost, and West Antarctica where ice shelves lost 5.5 % (-5,563.1 km$^2$) of their 2009 area. From 2009 to 2019 our observations show that the WAIS and AP experienced overall cumulative mass loss, whereas the AP, Ross, and Ronne

Filchner, saw cumulative ice mass growth (Fig. 4). Ice shelves along the West Antarctic Ice Sheet lost 150.2 Gt/yr of ice mass, with individual drainage basins including Pine Island, Thwaites and Abbot contributing the most ice loss. On the Antarctic Peninsula, ice shelves also lost a total mass of 103.5 Gt/yr over the last decade, contributing significantly to freshwater input into the ocean. Larger shelves such as Ross, Ronne, and Filchner, gained 261.6 Gt/yr of ice. In East Antarctica, Baudouin, Totten, Mertz, and Nansen were the only shelves to lose ice (-5.3, -3.9, -49.7 and -2.1 Gt/yr

respectively), however, the region as a whole gained 50.9 Gt/yr of ice from 2009-2019.

### 3.8 Steady State Calving Flux

In the absence of an observed measurement of ice loss from iceberg calving, previous studies have used the steady-state calving approximation to estimate the volume of ice lost through calving processes (Rignot et al., 2013; Liu et al., 2015). This method assumes that all ice flow through a fixed flux-gate, usually located near or at the last known calving front

position, is lost through iceberg calving (Rignot et al., 2013). We calculated the mass change from both the observed and steady-state calving flux methods for all 34 ice shelves in Antarctica to assess the impact of fully accounting for observed change over the last decade (Table 1).

We calculated the ice mass change using the observed calving flux by multiplying the area difference from 2009 to 2019 by the mean ice thickness (Fretwell et al., 2013) at the most inland calving front and the density of ice (ρ=0.9166 Gt/km$^3$). The

ice mass change using a steady-state assumption was estimated using a flux gate located at the most inland observed calving front position on each ice shelf since 2009, where the mean ice speed is extracted from the gate location (Mouginot et al., 2019). This is multiplied by the mean ice thickness (Fretwell et al., 2013), the length of the calving front, and the density of ice (Rignot et al., 2013) (Table. 1). To compare the different methods, we calculated the difference between the two numbers on all ice shelves within the study, where mass loss was observed on 18 ice shelves and mass gain was observed on 16.

Overall, the steady-state assumption will overestimate ice loss on ice shelves that are advancing, and underestimate ice loss on ice shelves that are retreating. The assumption also does not hold well for any irregular behaviour, such as ice shelves that have lost ice through large calving events. Our observations show that Antarctic ice shelves gained 660.6 Gt of ice mass

from 2009 to 2019, whereas the steady state approach would estimate ice loss of -20,028.1 Gt over the same period (Table 1). The steady state calving flux approximation is closest to the observations on the Antarctic Peninsula; however, the amount of ice loss is significantly overestimated in both west and east Antarctica and on all large ice shelves. This shows that time-variable observations of calving flux are essential for accurately quantify the timing and volume of ice shelf calving flux in Antarctica.

## 4 Conclusions

This study has generated a comprehensive dataset of change in ice shelf area on 34 Antarctica ice shelves over the last decade. Overall, ice shelves on the Antarctic Peninsula and West Antarctica lost areas of 6,692.5 km$^2$ and 5,563.1 km$^2$ respectively, while East Antarctic Ice Shelves gained 3,532.1 km$^2$ of ice and the large ice shelves of Ross, Ronne, and Filchner grew by 14,027.9 km$^2$ (total). This dataset is a high spatial resolution record of change from 2009 to 2019, which shows the regional differences in ice shelf calving behaviour and documents the frequency and magnitude of ice shelf calving events across the continent on decadal timescales. These observations will be useful for regional studies of ice shelf change in Antarctica and can be used as an input dataset for modelling studies or as a validation dataset for future studies that develop more automated methods of measuring change in ice shelf calving front position. Future studies should use the historical satellite data archives to extend the record of ice shelf area change which will allow us to establish whether there is a long-term change in ice shelf calving frequency in Antarctica. We must develop and apply automated techniques to increase the frequency with which calving front measurements can be made, particularly on smaller ice shelves and glaciers, which will allow shorter term, seasonal calving behaviour to be characterised and monitored.

## Data availability

All raw data can be provided by the corresponding authors upon request.

## Author contributions

JA and AH planned the research; JA and HS performed the measurements; JA and AH analyzed the data, wrote the manuscript draft, and reviewed and edited the manuscript.

## Competing interests

The authors declare that they have no conflict of interest.



**Acknowledgements**

This work was led by J.R.A at the University of Minnesota's Department of Soil, Water, and Climate and the School of Earth and Environment at the University of Leeds. J.R.A was supported by the Future Investigators in NASA Earth and Space Science and Technology (FINESST) Award. A.E.H was supported by the Natural Environment Research Council (NERC) DeCAdeS project (NE/T012757/1) and the ESA Polar+ Ice Shelves project (ESA-IPL-POE-EF-cb-LE-2019-834). The authors gratefully acknowledge the National Aeronautics and Space Administration, for acquiring the MODIS satellite data. We acknowledge the use of datasets produced through the NASA Measures programme and are grateful for funding the development of long-term climate data records from satellite observations.

**Supplementary Online Information**

See supplementary document.

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
