# Peer review of "Change in Antarctic Ice Shelf Area from 2009 to 2019"

_EGUsphere, 2022_

## Referee Comment (RC1)

***Change in Antarctic Ice Shelf Area from 2009 to 2019* by Andreasen et al.**

Reviewed by Chad A. Greene (NASA/JPL).

*Initial Reactions*

In this paper, Andreasen et al. describe a new, independently derived, annual, pan-Antarctic mapping of ice shelf calving fronts. The paper is very well written, the methods are sound and clearly described, and as the authors point out, the work of mapping and understanding calving is an important and timely endeavor. I am happy to see this effort, and I hope that with a few adjustments it will be published in *The Cryosphere*.

*Previous Work*

My main concern is that this paper is similar to a number of studies that are already in the literature, and although some of the previous work is acknowledged in the present manuscript, it's unclear how the new findings build on previous efforts. If the present study is not intended to explore new ground, that may in fact be okay, as there is real value in independent analysis that replicates established findings. But if the purpose of this study is only to replicate previous studies, then I'd like to see more clarity about which previous results are reaffirmed here, and who might've gotten it wrong in previous studies. If the present work finds any notable disagreement with previous studies, then I'd like to see that clearly stated and I'd like to see some discussion about why different groups might be coming up with different numbers, and what the differences might mean in a broader context.

A couple of Celia Baumhoer's papers are cited in this manuscript, but I'm afraid the most relevant one to the present study has been overlooked. In her 2021 paper, terminus positions were mapped for 1997, 2009, and 2018, and the paper investigated the environmental factors that led to terminus position changes during each epoch. The present manuscript presents effectively the second half of the time series from Baumhoer et al., 2021, but without looking into potential causes of terminus position change.

Some other work worth mentioning in the manuscript includes a pan-Antarctic survey of calving fronts by Miles et al., 2016, a recent regional study of calving fronts by Christie et al., 2022, and a 15-year annual pan-Antarctic calving dataset by Qi et al., 2021. Also, I'm not sure if it's citeable yet, but the authors may want to be aware of the high-resolution IceLines coastline dataset: https://download.geoservice.dlr.de/icelines/files/

*Main Findings*

As far as I can tell, the analysis is sound, the main findings are accurate, and everything generally agrees with the results of previous studies. It's somewhat tricky, however, to frame the results in a way that won't be easily misunderstood, particularly in this case, where changes over 10 years are dominated by just a few ice shelves whose calving cycles repeat every few decades. I am slightly concerned that a cursory glance at the abstract and conclusions might give the impression that Antarctica is in an overall phase of growth, when the present analysis

has only captured a small portion of the multi-decade calving cycles of the big ice shelves that dominate the continent-wide totals. My coauthors and I ran into this problem when we tried to describe a 24 year calving time series in a recent paper, and I'm not sure if we got the wording exactly right, but we did our best to put the results of our short time series into the context of the longer-term calving cycles of the big ice shelves. I'd like to see some more direct language or clauses in the abstract to make it clear that the authors are not implying that Antarctica is somehow already on track to recovery from climate change.

***Less Results, More Discussion***
The Results section is lengthy and presents a long list of numbers, most of which are already presented in Table, 1, and at times it's unclear why certain numbers are worth mentioning or how they change our understanding of ice shelf calving. An attempt has been made to provide context in the Results section, for example by mentioning the sea level potential of the Aurora Subglacial Basin in the same paragraph as the calving-front position change estimates for Totten, but no conceptual bridge is provided to link calving processes to the doomsday value of sea level potential. As a consequence, the Results section feels somewhat incohesive at times, and it's unclear how all the facts and figures are related to each other or which findings might be most significant.

I recommend significantly abbreviating the Results section, to put the main findings in clear focus. For anyone who wishes to know the exact amount of area change of a specific ice shelf between two arbitrary dates, I recommend sharing the data, so they can explore it as they see fit. Separately, the inclusion of a Discussion section may provide a better place to tell the "story" of a few key locations that may be of interest. Sticking with Totten as an example (but this is by no means a prod to focus on Totten in the revision), Sue Cook did some modeling work to understand the glaciological factors that can prime Totten for calving (Cook et al., 2018), Bertie Miles looked at environmental forcing and ice-front change there (Miles et al., 2016), and I've got a paper on Totten's dynamic sensitivity to calving (Greene et al., 2018). By following the thread of what causes calving to how calving impacts glacier dynamics, we gain a better understanding of how the present results are related to that 3.5 m sea level potential of the Aurora Subglacial Basin. Readers will appreciate this sort of "tying things together", as it will help us understand the importance of your results.

***Data Availability***
The real value of this paper is that it describes an independently derived calving-front dataset. The trouble is, the dataset apparently hasn't been placed in any public repository, it's not included as a supplement to the manuscript, and it's unclear if or how anyone will ever be able to access it, use it, build on this work, or directly evaluate the data. I do see a statement that the data will be made available upon request, but I think the field is trying to move beyond the old culture of sharing data via private handshake deals. (Sharing data "upon request" often fails when authors leave academia, and the social dynamic of needing to beg strangers for data tends to favor the well-connected and contribute to the Matthew Effect.) So that the data can be evaluated and we can feel confident that it will be made available to all, I'd like to see the data placed in a long-term data repository or uploaded as a supplement to this manuscript.

**Minor comments**

**Throughout:** Area change estimates are presented to 0.1 km$^2$ precision. That's probably a tad too precise, particularly given that uncertainty is stated as being 1 km$^2$.

**L7:** "50-years" hyphen is unnecessary.

**L15,16, and a few other places:** Only the word "Antarctic" needs to be capitalized in the phrase "Antarctic ice shelf" or "Antarctic ice shelves". I think we only capitalize "Ice Shelf" when it's part of the official name of a specific ice shelf.

**L51:** "there are only five examples of regional assessments that have been updated since 2011" The wording here might make some folks feel left out. I'm thinking of *Antarctic ice-shelf advance driven by anomalous atmospheric and sea-ice circulation* by Christie et al., 2022, *Environmental drivers of circum-Antarctic glacier and ice shelf front retreat over the last two decades* by Baumhoer et al., 2021, *Pan–ice-sheet glacier terminus change in East Antarctica reveals sensitivity of Wilkes Land to sea-ice changes* by Miles et al., 2016, and a handful of other studies that have looked at the histories of single ice shelves or neighboring ice shelves. Consider rewording the sentence to focus on the positive—Talk about the work that *has* been done, rather than the focusing on what *hasn't* been done.

**L53:** "In this study we address this gap…" It's not entirely clear what gap is being addressed. Consider wording more along the lines of, "In this study, we build on previous work to answer such-and-such remaining question" or "We build on previous work to gain a better understanding of such-and-such." (The "yes, and" rule of improv is often a good starting point for motivating scientific studies, and it always feels better than "yes, but".)

**L62:** Hyphenate "cloud-free".

**L83:** The method of quantifying uncertainty in terminus pick position sounds sensible to me. How does picking uncertainty propagate into uncertainty in final estimates of area and mass change?

**L97 and elsewhere:** "We computed the mean annual rate of calving by dividing the total area change by the number of years observed…" The pedant in me is reacting to this framing. Ice shelves may grow at a linear rate, and they may retreat at an ~linear rate when successive small calving events occur over many years, but in the case of a single calving event over the course of the observation period, it feels somewhat inappropriate to describe this as a rate of change. It's more appropriate, in my opinion, to talk about the cumulative change over the observation period, without dividing by time.

**L102:** How are uncertainties in ice thickness handled when estimating ice mass changes? Keeping in mind that Bedmap2 ice shelf thickness is estimated by subtracting modeled firn air content (order of 20 or 30 m) from surface elevation measurements and applying hydrostatic

inversion (multiply by 9.3), the firn correction alone can influence ice thickness by hundreds of meters, and firn is rather poorly constrained in Antarctica. I realize there's no good way to validate ice shelf thickness where it has not been directly measured (and even radar has its uncertainties), but it would be good to have some approximate bounds on the mass change estimates that are presented in this study. I recommend making some reasonable guess at thickness uncertainty, and propagate it into the mass change estimates.

**L106:** I'm not entirely sure I follow the logic of the ice shelf area uncertainty estimates. Above, the uncertainty in picking position is estimated at 254 m, and that sounds very reasonable to me. I interpret line 106 to mean that the 254 m value is not considered in the area uncertainty. Line 106 says accuracy is rounded to 1 $km^2$, but it's unclear whether the 1 $km^2$ uncertainty applies to each ice shelf separately, or Antarctica as a whole. My intuition says 1 $km^2$ may be a reasonable estimate of area uncertainty for a small ice shelf, but those 254 m position errors are likely highly correlated along the edge of the bigger ice shelves like the Ronne. The Ronne front is some ~2000 MODIS pixels wide, so a fully correlated 250 m picking error should result in something like 125 $km^2$ uncertainty for the Ronne, if I've done the math correctly. Perhaps errors are not fully correlated along the entire Ronne ice front, but I suspect the measurements are not accurate to 1 $km^2$ for the big ice shelves.

**L219:** I think "tsunami" can be uncapitalized.

**L291:** Units appear on this line as m/a, whereas in the rest of the manuscript it's m/yr. According to the style guide (https://www.the-cryosphere.net/submission.html) they should all be written exponentially (m $yr^{-1}$).

**L306:** The heading "Rapid Area Growth" strikes me as a little funny, given that it's occurring at a glacial pace. Perhaps "Steady Area Growth" would be a better descriptor? Feel free to disagree.

**L357:** Liu et al., 2015 is incorrectly cited as an example of a study that estimates steady-state calving flux. Similar to Qi et al., 2021, they actually just counted the icebergs that were bigger than 1 $km^2$ (and the uncounted icebergs smaller than that might be why their calving estimates are so much lower than Rignot's). If you'd like to cite another highly relevant paper that used steady-state analysis, check out Depoorter et al., 2013.

**Results:** Cook Ice Shelf drains a major marine-based subglacial basin, and the ice flow has been shown to be sensitive to changes in the terminus position (Jordan et al., 2022). Is there a reason Cook was excluded from this study?

**References**

Baumhoer, C.A., Dietz, A.J., Kneisel, C., Paeth, H. and Kuenzer, C., 2021. Environmental drivers of circum-Antarctic glacier and ice shelf front retreat over the last two decades. *The Cryosphere*, 15(5), pp.2357-2381.

Christie, F.D., Benham, T.J., Batchelor, C.L., Rack, W., Montelli, A. and Dowdeswell, J.A., 2022. Antarctic ice-shelf advance driven by anomalous atmospheric and sea-ice circulation. *Nature Geoscience*, 15(5), pp.356-362.

Cook, S., Åström, J., Zwinger, T., Galton-Fenzi, B.K., Greenbaum, J.S. and Coleman, R., 2018. Modelled fracture and calving on the Totten Ice Shelf. *The Cryosphere*, 12(7), pp.2401-2411.

Depoorter, M.A., Bamber, J.L., Griggs, J.A., Lenaerts, J.T., Ligtenberg, S.R., van den Broeke, M.R. and Moholdt, G., 2013. Calving fluxes and basal melt rates of Antarctic ice shelves. *Nature*, 502(7469), pp.89-92.

Greene, C.A., Young, D.A., Gwyther, D.E., Galton-Fenzi, B.K. and Blankenship, D.D., 2018. Seasonal dynamics of Totten Ice Shelf controlled by sea ice buttressing. *The Cryosphere*, 12(9), pp.2869-2882.

Jordan, J.R., Gudmundsson, G.H., Jenkins, A., Stokes, C.R., Miles, B.W. and Jamieson, S.S., 2022. The sensitivity of Cook Glacier, East Antarctica, to changes in ice-shelf extent and grounding-line position. *Journal of Glaciology*, 68(269), pp.473-485.

Miles, B.W., Stokes, C.R. and Jamieson, S.S., 2016. Pan–ice-sheet glacier terminus change in East Antarctica reveals sensitivity of Wilkes Land to sea-ice changes. *Science Advances*, 2(5), p.e1501350.

Qi, M., Liu, Y., Liu, J., Cheng, X., Lin, Y., Feng, Q., Shen, Q. and Yu, Z., 2021. A 15-year circum-Antarctic iceberg calving dataset derived from continuous satellite observations. *Earth System Science Data*, 13(9), pp.4583-4601.

---

## Author Response (AR1)

**Manuscript egusphere-2022-1087**
**Response to Reviewers**

We thank the reviewers and the editor for their time and effort in reviewing our paper, "Change in Antarctic Ice Shelf Area from 2009 to 2019", submitted for publication in The Cryosphere. We welcome the positive feedback and insightful comments which we have endeavored to fully address in this resubmitted revision, and we hope you agree this improves the manuscript. We have incorporated the majority of the suggestions made by the reviewer, and in the limited cases where we have not, we have provided a detailed description of the justification for each decision. The changes are highlighted in the manuscript through the track changes function. Please see below a point-by-point response to the reviewers' comments, where all line numbers refer to the revised manuscript file with the tracked changes.

| ID | Comment | Response |
|---|---|---|
| **Reviewer #1** | | |
| 1 | **Reviewer #1 (Remarks to the Author):**

 a. My main concern is that this paper is similar to a number of studies that are already in the literature, and although some of the previous work is acknowledged in the present manuscript, it's unclear how the new findings build on previous efforts. If the present study is not intended to explore new ground, that may in fact be okay, as there is real value in independent analysis that replicates established findings. But if the purpose of this study is only to replicate previous studies, then I'd like to see more clarity about which previous results are reaffirmed here, and who might've gotten it wrong in previous studies. If the present work finds any notable disagreement with previous studies, then I'd like to see that clearly stated and I'd like to see some discussion about why different groups might be coming up with different numbers, and what the differences might mean in a broader context.

 b. A couple of Celia Baumhoer's papers are cited in this manuscript, but I'm afraid the most relevant one to the present study has been overlooked. In her 2021 paper, terminus positions were mapped for 1997, 2009, and 2018, and the paper investigated the environmental factors that led to terminus position changes during each epoch. The present manuscript presents effectively the second half of the time series from Baumhoer et al., 2021, but without looking into potential causes of terminus position change.

 Some other work worth mentioning in the manuscript includes a pan-Antarctic survey of | Done.

 a. The purpose of our study is to assess area change across all major ice shelves in Antarctica from 2009 to 2019. At the time this work was completed calving front datasets were not routinely produced across the continent, and most change analysis in the literature was limited to regional case studies which we have cited in our introduction. More recent work, primarily Greene et al., (2022), does provide a continent-wide assessment of ice shelf area change, so our work is complimentary to this manuscript. We provide a direct quantitative comparison between these two continent wide estimates in a new table in the supplementary material (Table S3), and we provide a reason for any differences on an ice shelf by ice shelf basis. We note that as the acquisition date of the underlying satellite images is different for both studies, and the spatial resolution of the calving front product is also different, there will always be some minor differences between the two results. We hope that both datasets will be complimentary and of use to the scientific community. We think there is considerable value in producing an independent dataset and analysis of ice shelf migration over this relatively long 11-year period. We focus on documenting regional patterns of calving front change and try to categorise ice shelves into different types of calving behaviour which hasn't been done in previous studies. We discuss the change |

| | | |
|---|---|---|
| | calving fronts by Miles et al., 2016, a recent regional study of calving fronts by Christie et al., 2022, and a 15-year annual pan-Antarctic calving dataset by Qi et al., 2021. Also, I'm not sure if it's citeable yet, but the authors may want to be aware of the high-resolution IceLines coastline dataset: https://download.geoservice.dlr.de/icelines/files/ | observed in our study period with respect to other published literature, to provide wider context for why each ice shelf has changed.

**b.** We agree that understanding the environmental forces driving change in calving front position is both interesting and important, and the Baumhoer et al., 2021 study does an excellent job of this. In this study we focussed on the annual change of each individual ice shelf over a decade-long period, and we hope that in the future our new dataset will be used for studies of this type. |
| 2 | As far as I can tell, the analysis is sound, the main findings are accurate, and everything generally agrees with the results of previous studies. It's somewhat tricky, however, to frame the results in a way that won't be easily misunderstood, particularly in this case, where changes over 10 years are dominated by just a few ice shelves whose calving cycles repeat every few decades. I am slightly concerned that a cursory glance at the abstract and conclusions might give the impression that Antarctica is in an overall phase of growth, when the present analysis has only captured a small portion of the multi-decade calving cycles of the big ice shelves that dominate the continent-wide totals. My coauthors and I ran into this problem when we tried to describe a 24 year calving time series in a recent paper, and I'm not sure if we got the wording exactly right, but we did our best to put the results of our short time series into the context of the longer-term calving cycles of the big ice shelves. I'd like to see some more direct language or clauses in the abstract to make it clear that the authors are not implying that Antarctica is somehow already on track to recovery from climate change. | Comment. In addition to stating the overall Antarctic ice shelf area change number in the abstract we do also state in the abstract that there are clear regional differences, with retreat on the peninsula and WAIS, and advance in EAIS and on the large Ronne-Filchner and Ross ice shelves. We are also careful not to use language that might be misleading, like 'recovery'. There is clearly some nuance about how best to present these numbers within the length constraint of an abstract, but we do think that it's highly unlikely that a reader could conclude from our abstract that the results show recovery from climate change. |
| 3 | **a.** The Results section is lengthy and presents a long list of numbers, most of which are already presented in Table, 1, and at times it's unclear why certain numbers are worth mentioning or how they change our understanding of ice shelf calving. An attempt has been made to provide context in the Results section, for example by mentioning the sea level potential of the Aurora Subglacial Basin in the same paragraph as the calving-front position change estimates for Totten, but no conceptual bridge is provided to link calving processes to the | Done.
**a.** We use the results section to present a comprehensive description of change on each ice shelf and highlight important numbers that encompass each shelf's trend of growth/retreat over the 2009-2019 decade. The inclusion of key values allows the reader to better interpret the many numbers included in Table 1. We have included a simplified and more legible version of Table 1 in the main text, while including the full table in the |

| | | |
|---|---|---|
| | doomsday value of sea level potential. As a consequence, the Results section feels somewhat incohesive at times, and it's unclear how all the facts and figures are related to each other or which findings might be most significant. I recommend significantly abbreviating the Results section, to put the main findings in clear focus. | supplementary materials to improve readability (see Supplemental Table 2). We opt not to single out or highlight specific ice shelves over others because this is already done in regional case study papers. The patterns of retreat/growth within the 2009-2019 time period are of equal importance on all ice shelves, so it was interesting to describe many of the lesser studied ice shelves in this paper. We have used the results to categorise the 34 ice shelves into 6 different calving regimes, providing useful wider context into general patterns of behaviour. We demonstrate the value of measuring the observed change in calving flux, as opposed to the steady state assumption. |
| | **b.** For anyone who wishes to know the exact amount of area change of a specific ice shelf between two arbitrary dates, I recommend sharing the data, so they can explore it as they see fit. | |
| | **c.** Separately, the inclusion of a Discussion section may provide a better place to tell the "story" of a few key locations that may be of interest. Sticking with Totten as an example (but this is by no means a prod to focus on Totten in the revision), Sue Cook did some modeling work to understand the glaciological factors that can prime Totten for calving (Cook et al., 2018), Bertie Miles looked at environmental forcing and ice-front change there (Miles et al., 2016), and I've got a paper on Totten's dynamic sensitivity to calving (Greene et al., 2018). By following the thread of what causes calving to how calving impacts glacier dynamics, we gain a better understanding of how the present results are related to that 3.5 m sea level potential of the Aurora Subglacial Basin. Readers will appreciate this sort of "tying things together", as it will help us understand the importance of your results. | **b.** The data will of course be made freely available to the community. We are in the process of uploading the data to the opensource Pangea repository, and it will be freely available at the time of publication (see comment #4). **c.** See response 3a. |
| 4 | The real value of this paper is that it describes an independently derived calving-front dataset. The trouble is, the dataset apparently hasn't been placed in any public repository, it's not included as a supplement to the manuscript, and it's unclear if or how anyone will ever be able to access it, use it, build on this work, or directly evaluate the data. I do see a statement that the data will be made available upon request, but I think the field is trying to move beyond the old culture of sharing data via private handshake deals. (Sharing data "upon request" often fails when authors leave academia, and the social dynamic of needing to beg strangers for data tends to favor the well-connected and contribute to the Matthew Effect.) So that the data can be evaluated and we can feel | Done. See response 3b. The data will of course be made freely available to the community. All calving front shapefiles are currently in the process of being publicly available on PANGAEA after the paper is out of Pre-Print and has gone through full peer-review. Edit line 388: "The 2009-2019 MODIS calving front data that support the findings of this study will be available from PANGAEA." |

| | | |
|---|---|---|
| | confident that it will be made available to all, I'd like to see the data placed in a long-term data repository or uploaded as a supplement to this manuscript. | |
| 5 | Throughout: Area change estimates are presented to 0.1 km2 precision. That's probably a tad too precise, particularly given that uncertainty is stated as being 1 km2. | Done. We agree with this suggestion and have removed the decimal point precision to ensure that we are not misrepresenting uncertainties. |
| 6 | L7: "50-years" hyphen is unnecessary. | Done.
Edit line 7: "50 years" |
| 7 | L15,16, and a few other places: Only the word "Antarctic" needs to be capitalized in the phrase "Antarctic ice shelf" or "Antarctic ice shelves". I think we only capitalize "Ice Shelf" when it's part of the official name of a specific ice shelf. | Done.
Edit lines 6, 14, 15, 24, 377: "Antarctic ice shelf..." or "Antarctic ice shelves..." |
| 8 | L51: "there are only five examples of regional assessments that have been updated since 2011" The wording here might make some folks feel left out. I'm thinking of Antarctic ice-shelf advance driven by anomalous atmospheric and sea-ice circulation by Christie et al., 2022, Environmental drivers of circum-Antarctic glacier and ice shelf front retreat over the last two decades by Baumhoer et al., 2021, Pan–ice-sheet glacier terminus change in East Antarctica reveals sensitivity of Wilkes Land to sea-ice changes by Miles et al., 2016, and a handful of other studies that have looked at the histories of single ice shelves or neighboring ice shelves.
Consider rewording the sentence to focus on the positive—Talk about the work that has been done, rather than the focusing on what hasn't been done. | Done. Several of the studies mentioned were only just published while this manuscript was in its final stages of preparation. We have made the following edits following the reviewers' useful suggestions.

Edit Line 48-51: "Due to the importance of this glaciological parameter, there are several recent publications that measure change in Antarctic ice shelf calving front location, from regional assessments to full continent-wide evaluations..."
Edit Line 51-52: "In this study, we expand on this previous work and provide a Circum-Antarctic survey by mapping the annual calving..."
Edit Lines 53-54: "The results provide a comprehensive assessment of ice front migration across Antarctica over the last decade, expanding on historic patterns of ice movement and enabling areas of growth and..." |
| 9 | L53: "In this study we address this gap..." It's not entirely clear what gap is being addressed. Consider wording more along the lines of, "In this study, we build on previous work to answer such-and-such remaining question" or "We build on previous work to gain a better understanding of such-and-such." (The "yes, and" rule of improv is often a good starting point for motivating scientific studies, and it always feels better than "yes, but".) | Done. See comment 8 response. Text edited to reflect comment. |
| 10 | L62: Hyphenate "cloud-free". | Done. Edit Line 60: "Cloud-free" |
| 11 | L83: The method of quantifying uncertainty in terminus pick position sounds sensible to me. How does picking uncertainty propagate into uncertainty in final estimates of area and mass change? | Comment. Please see additional details in response to comment #14. |
| 12 | L97 and elsewhere: "We computed the mean annual rate of calving by dividing the total area | Done. Calculating the mean annual rate provides helpful context for ice shelves that |

| | | |
|---|---|---|
| | change by the number of years observed...” The pedant in me is reacting to this framing. Ice shelves may grow at a linear rate, and they may retreat at an ~linear rate when successive small calving events occur over many years, but in the case of a single calving event over the course of the observation period, it feels somewhat inappropriate to describe this as a rate of change. It's more appropriate, in my opinion, to talk about the cumulative change over the observation period, without dividing by time. | are steadily retreating and advancing (sections 3.3 and 3.6); however, the reviewer is correct that this metric is less representative for ice shelves that have undergone major calving events where the overall ice loss is not indicative of a steady rate of change. We placed the mean annual rate of calving for ice shelves that experienced major calving events in brackets in Table 1, to highlight this point to the reader.

Edit Table 1 and Table caption: “Table 1: Summary table with data on each ice shelf including: area change from 2009 to 2019, the absolute difference, percentage difference, and rate of change between the first and last recorded dates (ice shelves that have experienced major calving events are indicated with brackets)...” |
| 13 | L102: How are uncertainties in ice thickness handled when estimating ice mass changes? Keeping in mind that Bedmap2 ice shelf thickness is estimated by subtracting modeled firn air content (order of 20 or 30 m) from surface elevation measurements and applying hydrostatic inversion (multiply by 9.3), the firn correction alone can influence ice thickness by hundreds of meters, and firn is rather poorly constrained in Antarctica. I realize there's no good way to validate ice shelf thickness where it has not been directly measured (and even radar has its uncertainties), but it would be good to have some approximate bounds on the mass change estimates that are presented in this study. I recommend making some reasonable guess at thickness uncertainty, and propagate it into the mass change estimates. | Comment. As the reviewer points out, ice thickness estimates may carry large uncertainties which vary spatially. In this study we use the Bedmap2 ice thickness to calculate both the steady state and observed calving flux, so any difference can be attributed to the change in calving measurement alone. We don't account for the uncertainty in the thickness data in our results. As we are making the calving front dataset freely available to the community, all results will be directly reproducible from the same datasets, and colleagues can use their preferred ice thickness when doing any further analysis. |
| 14 | L106: I'm not entirely sure I follow the logic of the ice shelf area uncertainty estimates. Above, the uncertainty in picking position is estimated at 254 m, and that sounds very reasonable to me. I interpret line 106 to mean that the 254 m value is not considered in the area uncertainty. Line 106 says accuracy is rounded to 1 km2, but it's unclear whether the 1 km2 uncertainty applies to each ice shelf separately, or Antarctica as a whole. My intuition says 1 km2 may be a reasonable estimate of area uncertainty for a small ice shelf, but those 254 m position errors are likely highly correlated along the edge of the bigger ice shelves like the Ronne. The Ronne front is some ~2000 MODIS pixels wide, so a fully correlated 250 m picking error should result in something like 125 km2 uncertainty for the Ronne, if I've done the math correctly. Perhaps errors are not | Done. It was important to characterize the uncertainty on the calving front location measurement, and we chose to do this by testing how accurate the manual delineation was on Dotson Ice shelf. This provides us with the 254 m number, which we do think is a good indication of the uncertainty on our core measurement. It wasn't feasible to repeat this analysis on all ice shelves due to the time-consuming nature of the method used, even though we fully acknowledge that there is inevitably regional variability on the quality of the measurements. For example, Dotson and Ronne have clear, cloudless MODIS imagery as well as relatively straight and easy-to-navigate fronts, reducing potential delineation error. However, more complex shelves, such as Shackleton, have intricate calving fronts |

| | | |
|---|---|---|
| | fully correlated along the entire Ronne ice front, but I suspect the measurements are not accurate to 1 km2 for the big ice shelves. | with crevassing, sea ice, and the presence of cloud cover, making the margin for error much higher. More automated methods of generating calving front datasets will be much better placed to provide a spatially and temporally variable error estimate.

Lastly, we round our areas to 1 km$^2$ precision based on methodology found in Cook and Vaughan's 2010 publication. This correction can be found in Table 1 and Sup. Table 2.

Edit Lines 105-106: "...in line with the methodology of previous studies (Cook and Vaughan, 2010) as well as to account for errors within the calving front delineation (254 m)." |
| 15 | L219: I think "tsunami" can be uncapitalized. | Done.

Edit line 216: "tsunami" |
| 16 | L291: Units appear on this line as m/a, whereas in the rest of the manuscript it's m/yr. According to the style guide (https://www.the-cryosphere.net/submission.html) they should all be written exponentially (m yr-1). | Done.

Edit lines 14, 148, 178, 180, 197, 207, 218, 220, 225, 244, 269, 274, 296, 308, 315, 320, 325: "m yr$^{-1}$", "km$^2$ yr$^{-1}$", and "km yr$^{-1}$" |
| 17 | L306: The heading "Rapid Area Growth" strikes me as a little funny, given that it's occurring at a glacial pace. Perhaps "Steady Area Growth" would be a better descriptor? Feel free to disagree. | Done. "Rapid Area Growth" has been renamed to be "Rapid Area Advance" to match other usage of the term "advance" and is an appropriate descriptor when comparing the growth of these glaciers to the speed at which the other glaciers are growing/receding. This is because the area is rapidly advancing but not at a steady annual pace (see Fig. 3e). This is a stark contrast from the "Steady Calving Front Advance" category, which describes calving fronts that are growing at a slower but steady annual rate (see Fig. 3f).

Edit Lines 118-119, 182, 251, 253, 254, 256, 269, 284: "rapid calving front advance" |
| 18 | L357: Liu et al., 2015 is incorrectly cited as an example of a study that estimates steady-state calving flux. Similar to Qi et al., 2021, they actually just counted the icebergs that were bigger than 1 km2 (and the uncounted icebergs smaller than that might be why their calving estimates are so much lower than Rignot's). If you'd like to cite another highly relevant paper that used steady-state analysis, check out Depoorter et al., 2013. | Done. We agree that Liu et al., 2015 does not utilize steady-state calving flux calculations. We cited this paper in this location because it explicitly discusses the importance of avoiding using this assumption. We have clarified this in the text to avoid confusion, and we also cite Depoorter et al., (2013) as the reviewer suggests.

Edit line 352: "Depoorter et al., 2013"
Edit line 371-373: "These comparisons are in agreement with past studies that compare observed data to steady state (Liu et al., 2015) and show..." |

| 19 | Results: Cook Ice Shelf drains a major marine-based subglacial basin, and the ice flow has been shown to be sensitive to changes in the terminus position (Jordan et al., 2022). Is there a reason Cook was excluded from this study? | Comment. Cook is a really interesting ice shelf and there are lots of papers documenting its importance. As you might imagine it was a significant task to manually delineate the ice shelf calving fronts on the 34 ice shelves we did include in our study, so it was simply a function of time that prevented us from extending the scope further. |

| ID | Comment | Response |
|---|---|---|
| **Reviewer #2** | | |
| 20 | **Reviewer #2 (Remarks to the Author):**
L51. I would not call ice shelf calving front position rare given the dense amount of paper that came out just in 2022. Maybe you can stress out that there is only one time series that goes far back in time (Green et al., 2022)? | Done. We have edited the text to reflect the reviewer's comment. While similar studies have recently been published, the data collected for this manuscript was completed prior to the publication of Greene et al. (2022).

Edit Lines 48-51: "Due to the importance of this glaciological parameter, there are several publications that measure change in Antarctic ice shelf calving front location, from regional assessments to full continental evaluations (MacGregor et al., 2012; Lilien et al., 2018; Wuite et al., 2019; Baumhoer et al., 2018, 2019, 2021; Greene et al., 2022; Christie et al., 2022)."
Edit Line 51-52: "In this study, we expand on this previous work and provide a Circum-Antarctic survey by mapping the annual calving..."
Edit Lines 53-54: "The results provide a comprehensive assessment of ice front migration across Antarctica over the last decade, expanding on historic patterns of ice movement and enabling areas of growth and..." |
| 21 | L53. Another thing that could be stressed out is that recent studies are combining different sources of data (MOA, RAMP, MODIS, Sentinel-1a). Those datasets have different spatial resolution, with different related uncertainties. For example, the RAMP MOA and MODIS are composite mosaics, hence there is an uncertainties on seasonal front variations at these times right? What are these datasets most representative of? What are the uncertainties related to these datasets and the way they are combined? **One of the good points of this study is that the product is higher resolution and uses one single source of data, hence reducing sources of errors.**

Also from the Greene et al paper, we can see that the delineation of the coastlines is really rough, and lots of the fronts looks like staircase, and does not follow smoothly the ice frontal position. I am wondering, overall, how these "wrong" or "low resolution" delineation are impacting the total change in area. These points of comparison should be stressed out in this paper, to try to assess the quality of product, and how the ones from Andreasen should be considered as a reference compared to other studies (see figure below from Pine Island glacier). A comparison with products | Done. We thank the reviewer for these comments and agree that this study is unique in that it uses a consistent/single source of data to calculate ice shelf changes over an 11-year time span. The spatial resolution of the various satellites the reviewer highlights are different, and it would be interesting to sensitivity test the impact of this on the locations measured in future studies that take a multi-sensor approach. Certainly, the error estimate that we calculated in this study is directly related to the spatial resolution of the MODIS imagery, so a new error estimate should be calculated when measuring the calving front in different resolution satellite datasets.

We have updated the manuscript to provide a direct comparison of our measurements and the Greene et al (2022) result. Baumhoer et al. (2018) is also an extremely valuable dataset and we are sure the community will make use of all of these in future studies. It was out of the scope of this paper to do a formal intercomparison of all calving front datasets. |

| | | |
|---|---|---|
| | from Baumhoer et al., 2018 should also be performed, as it was derived from deep learning vs manual in this study.
 | Edit: To address the differences between Greene et al.'s (2022) calving fronts and ours, we have created a direct comparison of areas in a Supplementary Table with reasoning as to why the datasets differ (see Supplementary Table 3). |
| 22 | L58. Why not doing all ice shelves? If you are missing 20% then you can't have a title saying "Change in Antarctic Ice shelf Area", this is misleading. How much work would be needed to add those missing ice shelves? I think that you should really consider having this comprehensive view of ice shelves here, which will contribute in imposing this dataset as a reference. | Comment. Depending on your definition, Antarctica has around 300 ice shelves in total, many of which are small and do not account for a large proportion of the ice shelf area change on the continent. In this manuscript, we focused on the largest ice shelves first, and prioritized measuring the calving front at annual resolution over a decade, rather than measuring more ice shelves less frequently. |
| 23 | L61. Does the choice of the month will impact the results compared to Green et al ? I think they have chosen March right? | Done. We selected images based on the availability of MODIS satellite data (accounting for cloud cover), as well as considering the presence of sea ice which makes it more challenging to identify the calving front boundary. There is definitely seasonal variability in the calving front location in some regions, so the temporal sampling does matter. Studies in the future that use all-season and all-weather instruments such as synthetic aperture radar data, will be well placed to measure this short-term variability. To minimize the impact of any seasonal variability we took care to select images primarily from January and February (occasionally March if absolutely necessary) to provide the best conditions for digitizing the front while avoiding any seasonal bias. This is discussed in the data and methods section of the paper. |
| 24 | Figure 1. If you use a classification on the type of retreat, I would recommend using different symbol for the retreat types (circle, triangle, square…). Or some kind of symbology that would give an idea on the behavior of each ice shelf ? | Done. We really liked this suggestion and thank the reviewer for their comment. We wanted to retain the circle symbol on the current figure one as the diameter corresponds to the area change, so we felt that using different symbols might make interpretation of that information more challenging. We tried changing the outline of the circle to a color that corresponded to the calving regime, but this didn't look satisfactory. We have therefore added a new figure to the supplementary information file (Supplementary Figure 1) highlighting the |

| | | calving regime of the ice shelves with a symbol as suggested.

Edit Lines 253, 286: Additionally, the creation of this figure inspired a reorganization of the sections in this paper, with the new format placing the "Rapid Calving Front Advance" section (3.5) before the "Steady Calving Front Advance" section (3.6). As well changing the order of sub-images in figures 2 and 3 (switching the placement of Figures 3e. and 3f. as well as Figures 4e. and 4f). |
|---|---|---|
| 25 | L78. How does the sampling distance influences the accuracy of the ice front position and the overall derived ice shelf area ? Can you provide a figure in supplementary maybe, that shows how the ice shelf area change with the sampling ? That would be a good point of comparison with Green et al., who seem to have used a rather coarce sampling method. | Comment. The sampling resolution will impact ice shelves that have complex ice fronts most, as well as smaller ice shelves. Ice shelves with long relatively straight ice fronts (e.g. Ronne-Filchner or Ross) will be much less affected by sampling density as their calving front is simple and well represented by a line. We chose the sampling distance (points plotted every 1,000 meters) based on the resolution of the MODIS satellite images used (which have a pixel size of 250 x 250 meters). Although it is possible to down sample the spatial resolution of our calving fronts to the underlying MODIS imagery, we didn't sensitivity test the impact of this within this study. As suggested in response to reviewer comment 21, this will be a much more important consideration in multi-sensor studies when the underlying datasets are not all a consistent spatial resolution. |
| 26 | L93. Why not using the continent wide grounding line mapping made by Rignot et al to have the most accurate delineation ? In the Antarctic peninsula, updated grounding line position where also made using Sentinel-1, and could also be used to update grounding line with the yearly front position (Christie et al., 2022). | Done. We used the MEaSUREs grounding line dataset (Rignot et al., 2016), and we have clarified this in the text.

Edit Lines 91-92: "...MEaSUREs Antarctic Grounding Line from Differential Satellite Radar Interferometry, Version 2 (Rignot et al., 2016)..." |
| 27 | L100. Why using BEDMAP-2? How is the ice shelf thickness determined in there ? Why not using BedMachine that used REMA as a DEM ? | Done. See response to reviewer comment 13. |
| 28 | L106. Where does this 1 km2 comes from ? How was it calculated ? | Done. We round our areas to 1 $km^2$ precision based on methodology found in Cook and Vaughan's 2010 publication and accounting for errors within the calving front delineation (of 254 m).

Edit Lines 105-106: "...in line with the methodology of previous studies (Cook and Vaughan, 2010) as well as to account for errors within the calving front delineation (254 m)." |

| | | |
|---|---|---|
| 29 | Table 1. I am thinking that this Table should be given as a supplementary file. Here it is not realy readable. | Done. We have edited Table 1 to contain less information, and we have moved the full detailed table into the supplementary data (Supplementary Table 2) for those who wish to explore the data in more detail.

Edit Table 1: remove columns "Most Inland Calving Front (yr)", "Mean Ice Thickness (km yr$^{-1}$)", "Mean Ice Speed (km yr$^{-1}$)", and "Inland CFL Length (km)" |
| 30 | L124. This is over a short period I guess ? I think you need to include here a time variable for the definition of major calving events | Done.

Edit Line 123: "...over a short time period (calving events that occurred in less than a month)." |
| 31 | L159-160. I don't understand why this is not resolved in this study ? For those specific cases, can you investigate that using Sentinel-2 if MODIS is not sufficient ? | Done. This study focused on providing annual evaluations for each ice shelf extent due to MODIS being a passive sensor and therefore wouldn't collect sufficient data quality outside of the Austral Summer due to sea ice, increased cloud cover and lack of solar radiation. For specific large calving events, we provide information on the timing of each event because it is of particular interest to the community in Section 3.1. Additionally, we emphasize the importance of annual analysis as a baseline for future seasonal studies.

Edit Lines 156-157: "This analysis of calving events on an annual scale provides robust data for future studies to assess..." |
| 32 | L163-164. General comment for all the classification part: you need to give the reader some insights on how you determined those % area loss or increase used for the classification. Now it seems a bit random. | Done. We defined the percentage change based on the area change for each ice shelf during the study period. The grouping of ice shelves within each percentage threshold category helps us understand the wide range of behaviours in all regions of Antarctica. For some behaviour types, such as the large calving events, the classification is dependent on the time period, i.e., when the calving event took place. Studies over longer multi-decadal time periods would help better define each ice shelf's ice cycles and patterns of change; however, this dataset is a useful starting point for better understanding ice shelf area change in Antarctica.

We have included references to the time periods for each calving regime in the first sentence of each section (3.1-3.6).
Edit Line 123: "...over a short time period (calving events that occurred in less than a month)." |

| | | Edit Line 161-162: "...significant ice loss throughout the 11-year study period (2009-2019), loosing at least 15 % of their total area" Edit Line 186-187: "... lost less than 4 % of their total area over the 11-year study period (2009-2019)" Edit Line 229-230: "...but also have individual years of retreat within the last decade (2009-2019)" Edit Lines 254-255: "...by over 5 % during the 11-year study period (2009-2019)" Edit Line 287: "... ice shelves that have gradually grown in area from 2009-2019" |
|---|---|---|
| 33 | Section 3.7. Please compare your values with Green et al over the same time period | Done. We agree that a comparison in values would provide helpful insight on how the datasets compare. Please see comment #21 and Supplementary Table 3. |
| 34 | L366. Why do you use the most inland observed calving front position and not the latest 2019 position ? Why not the average ice front position over the time period ? | Done.

Edit Lines 362-364: "We used the most inland calving front position when calculating ice thickness and velocity to ensure that the fronts were within the spatial coverage of the thickness and velocity datasets." |
| 35 | L366. which ice speed do you use? did you make sure it was representative of the ice front date? | Done. We used MEaSUREs InSAR-based Antarctica Ice Velocity Map, version 2., with a 450 m resolution. This dataset is assembled from multiple satellite interferometric synthetic-aperture radar systems and was largely acquired during the International Polar Year 2007 to 2009, as well as between 2013 and 2016. This range represents a similar timeframe to that of the ice front dates.

Edit Lines 360: "...where the mean ice speed, MEaSUREs ice velocity at 450m resolution..." |
| 36 | L368. Split this sentence in two. | Done.

Edit Lines 364-365: "To compare the different methods, we calculated the difference between the two numbers on all ice shelves within the study. We observed mass loss on 18 ice shelves and mass gain on 16." |
| 37 | L370. Could you consider calculating a yearly calving flux? Would it make sense to compare it with the yearly mass losses from your changes in ice shelf area? | Comment. This is a good suggestion, and we hope our dataset will be used for this in future studies. |